# Kernelized Bayesian Softmax for Text Generation

**Ning Miao   Hao Zhou   Chengqi Zhao   Wenxian Shi   Lei Li**
ByteDance AI lab
{miaoning,zhouhao.nlp,zhaochengqi.d,shiwenxian,lileilab}@bytedance.com

## Abstract

Neural models for text generation require a softmax layer with proper word embeddings during the decoding phase. Most existing approaches adopt single point embedding for each word. However, a word may have multiple senses according to different context, some of which might be distinct. In this paper, we propose KerBS, a novel approach for learning better embeddings for text generation. KerBS embodies two advantages: *a)* it employs a Bayesian composition of embeddings for words with multiple senses; *b)* it is adaptive to semantic variances of words and robust to rare sentence context by imposing learned kernels to capture the closeness of words (senses) in the embedding space. Empirical studies show that KerBS significantly boosts the performance of several text generation tasks.

## 1   Introduction

Text generation has been significantly improved with deep learning approaches in tasks such as language modeling [Bengio et al., 2003, Mikolov et al., 2010], machine translation [Sutskever et al., 2014, Bahdanau et al., 2015, Vaswani et al., 2017], and dialog generation [Sordoni et al., 2015]. All these models include a softmax final layer to yield words. The softmax layer takes a context state ($h$) from an upstream network such as RNN cells as the input, and transforms $h$ into the word probability with a linear projection ($W \cdot h$) and an exponential activation. Each row of $W$ can be viewed as the embedding of a word. Essentially, softmax conducts embedding matching with inner-product scoring between a calculated context vector $h$ and word embeddings $W$ in the vocabulary.

The above commonly adopted setting for softmax imposes a strong hypothesis on the embedding space — it assumes that each word corresponds to a single vector and the context vector $h$ from the decoding network must be indiscriminately close to the desired word embedding vector in certain distance metric. We discover that such an assumption does not coincide with practical cases. Fig. 1 visualizes examples of the context vectors for utterances containing the examined words, calculated from the BERT model Devlin et al. [2019]. We make three interesting observations. *a)* Multi-sense: Not every word's context vectors form a single cluster. There are words with multiple clusters (Fig. 1b). *b)* Varying-variance: The variances of context vectors vary significantly across clusters. Some words correspond to smaller variances while others to larger variances (Fig. 1c). *c)* Robustness: There are outliers in the context space (Fig. 1b). These observations explain the ineffectiveness during training with the traditional softmax. The traditional way brings word embedding $W$ ill-centered with all context vectors of the same word – even though they might belong to multiple clusters. At the same time, the variances of different words are completely ignored in the plain softmax with inner-product as the similarity score. It is also vulnerable to outliers since a single anomaly would lead the word embedding to be far from the main cluster. In short, the softmax layer doesn't have sufficient expressiveness capacity.

Yang et al. [2018] propose Mixture-of-Softmax (MoS) to enhance the expressiveness of softmax. It replaces a single softmax layer with a weighted average of $M$ softmax layers. However, all words share the same fixed number of components $M$ and averaging weights, which heavily restrict MoS's capacity. Furthermore, the variances of context vectors are not taken into the consideration.

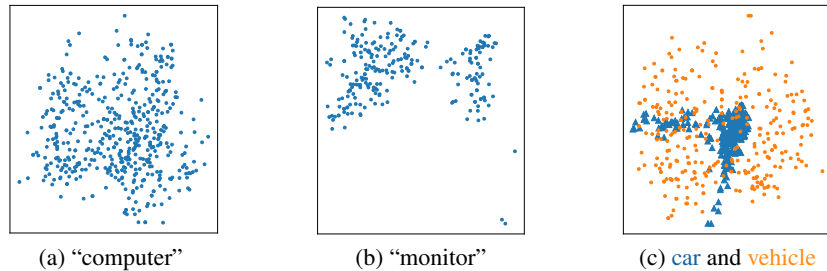

|        |        |        |
|:------:|:------:|:------:|
| (a) "computer" | (b) "monitor" | (c) car and vehicle |

Figure 1: Context vectors $h$ calculated from BERT and projected using PCA. Each point corresponds to one utterance containing the word. (a) "computer" has only one cluster; (b) "monitor" has two clusters, representing the verb (left) and the noun (right). The outlier at the lower right only appears in phrase "christian science monitor"; (c) "car" has smaller variance than "vehicle".

In this paper, we propose KerBS, a novel approach to learn text embedding for generation. KerBS avoids the above softmax issues by introducing a Bayesian composition of multiple embeddings and a learnable kernel to measure the similarities among embeddings. Instead of a single embedding, KerBS explicitly represents a word with a weighted combination of multiple embeddings – each is regarded as a "sense"[1]. The number of embeddings is automatically learned from the corpus as well. We design a family of kernel functions to replace the embedding matching (i.e. the matrix-vector dot-product) in softmax layer. With parameters learned from text, each word (or "sense") can enjoy individual variance in its embedding space. In addition, the kernel family is more robust to outliers than Gaussian kernels.

We conduct experiments on a variety of text generation tasks including machine translation, language modeling, and dialog generation. The empirical results verify the effectiveness of KerBS. Ablation study indicates that each part of KerBS, including the Bayesian composition and the kernel function, is necessary for the performance improvement. We also find that words with more semantic meanings are allocated with more sense embeddings, which adheres to our intuition.

## 2 Related work

**Word Embeddings.** Word2Vec [Mikolov et al., 2013] and GloVe [Pennington et al., 2014] learn distributed word representations from corpus in an unsupervised way. However, only one embedding is assigned to each word, which not only suffers from ignoring polysemy but also could not provide context related word embeddings. Recent works [Alec Radford and Sutskever, 2018, Peters et al., 2018, Devlin et al., 2019] indicates that pre-trained contextualized word representations are beneficial for downstream natural language processing tasks. BERT [Devlin et al., 2019] pre-train a masked language model with a deep bidirectional Transformer and it achieves state-of-the-art performance in various NLP tasks.

**Multi-Sense Word Embeddings.** Early works obtain multi-sense word embeddings by first training single point word embeddings and then clustering the context embeddings (for example, the average embedding of neighbor words). But these methods are not scalable and take lots of efforts in parameter tuning [Reisinger and Mooney, 2010, Huang et al., 2012]. Tian et al. [2014] introduce a probabilistic model, which uses a variable to control sense selection of each word. Liu et al. [2015] add a topic variable for each word, and condition word embeddings on the topic variable. Both of Tian et al. [2014] and Liu et al. [2015] can be easily integrated into Skip-Gram model [Mikolov et al., 2013], which is highly efficient. Other works [Chen et al., 2014, Jauhar et al., 2015, Chen et al., 2015, Wu and Giles, 2015] further improve the performance of multi-sense embeddings by making use of huge corpora such as WordNet Miller [1995] and Wikipedia. However, these works are mainly focused on text understanding rather than text generation.

**Word Embedding as a Distribution.** In order to represent the semantic breadth of each word, Vilnis and McCallum [2015] propose to map each word into a Gaussian distribution in the embedding

space. Instead of using cosine similarity in Mikolov et al. [2013], Vilnis and McCallum [2015] use KL-divergence of the embedding distributions to measure the similarities between words. To improve the numerical stability of Gaussian word embeddings, especially when comparing very close or very distant distributions, Sun et al. [2018] propose to replace KL-divergence with Wasserstein distance. Though Gaussian word embeddings perform well in word-level tasks such as similarity and entailment detection, they cannot be directly applied to the scenario of text generation, because it is difficult to perform embedding matching between Gaussian word embeddings and output embeddings, which are usually single points in the embedding space.

## 3 Background

Most text generation models generate words through an embedding matching procedure. Intuitively, at each step, upstream networks such as RNN decoders compute a *context vector* $h$ according to the encoded information from input and previously generated words. The context vector $h$ serves as a query to search for the most similar match from a pre-calculated vocabulary embeddings $W$. In practice, this is implemented with an inner-product between $W$ and $h$. Normalized probabilities over all words are computed with the softmax function. Words with the highest probabilities will be chosen during the inference process.

Specifically, given an utterance $\hat{y}$, a GRU decoder calculates as follows:

$$e_t = \text{LOOKUP}(W_{in}, \hat{y}_{t-1}), \tag{1}$$

$$h_t = \text{GRU}(h_{t-1}, e_t), \tag{2}$$

$$P(y_t = i) = \text{SOFTMAX}(h_t W)_i. \tag{3}$$

At time step $t$, its word embedding $e_t$ is obtained by looking up the previous output word in the *word embedding* matrix $W_{in} = [\bar{w}_1, \bar{w}_2, ..., \bar{w}_V]$ (Eq. (1)). Here $\bar{w}_i$ is the embedding of the $i$-th word in the vocabulary. $V$ is the vocabulary size. The context embedding $h_t$ of the $t$-th step will be obtained from GRU by combining information of $h_{t-1}$ and $e_t$ ( Eq. (2)). Other decoders such as Transformer Vaswani et al. [2017] work similarly.

Eq. (3) performs embedding matching between $h_t$ and $W$, and probabilities of words will be obtained by a softmax activation. Intuitively, to generate the correct word $\hat{y}_t$, the context embedding $h_t$ should lie in a small neighborhood around $\hat{y}_t$'s word embedding $w_{\hat{y}_t}$.

## 4 Proposed KerBS

In this section, we first introduce KerBS for text generation. It is designed according to the three observations mentioned in the introduction: multi-sense, varying-variance, and robustness. Then we provide a training scheme to dynamically allocate senses since it is difficult to directly learn the number of senses of each word.

### 4.1 Model Structure

KerBS assumes that the space of context vectors for the same word consists of several geometrically separate components. Each component represents a "sense", with its own variance. To better model their distribution, we replace Eq. (3) with the following equations:

$$P(y_t = i) = \sum_{j \in 0, 1, ..., M_i} P(s_t = \langle i, j \rangle). \tag{4}$$

Here, $s_t$ is the sense index of the step $t$. Its value takes $\langle i, j \rangle$ corresponding to the $j$-th sense of the $i$-th word in vocabulary. $M_i$ is the number of senses for word $i$, which may be different for different words. Instead of directly calculating the probabilities of words, KerBS first calculates the probabilities of all senses belonging to a word and then sums them up to get the word probability.

$$P(s_t = \langle i, j \rangle) = \text{Softmax}\big([\mathcal{K}_\theta(h_t, W)]\big)_i^j = \frac{\exp(\mathcal{K}_{\theta_i^j}(h_t, w_i^j))}{\sum_k \sum_{r \in 0, 1, ..., M_k} \exp(\mathcal{K}_{\theta_k^r}(h_t, w_k^r))}. \tag{5}$$

The probability of output sense $s_t$ in Eq. (5) is not a strict Gaussian posterior, as the training of Gaussian models in high dimensional space is numerical instable. Instead, we propose to use a carefully designed kernel function, to model the distribution variance of each sense. Concretely, we replace the inner product in Eq. (3) with kernel function $\mathcal{K}$, which depends on a variance-related parameter $\theta$. $[\mathcal{K}_\theta(h_t, W)]$ is a simplified notation containing all pairs of kernal values $\mathcal{K}_{\theta_i^j}(h_t, w_i^j)$. With different $\theta_i^j$ for each sense, we can model the variances of their distributions separately.

### 4.1.1 Bayesian Composition of Embeddings

In this part, we introduce in detail how KerBS models the multi-sense property of words. Intuitively, we use Bayesian composition of embeddings in text generation, because the same word can have totally different meanings. For words with more than one sense, their corresponding context vectors can be usually divided into separate clusters (see Figure 1). If we use single-embedding models such as traditional softmax to fit these clusters, the word embedding will converge to the mean of these clusters and could be distant from all of them. This may lead to poor performance in text generation.

As shown in Eq. (4), we can allocate different embeddings for each sense. We first obtain the sense probabilities by performing weight matching between context vector $h$ and sense embedding matrix $W$. Then we add up the sense probabilities belonging to each word to get word probabilities.

We adopt weight tying scheme [Inan et al., 2017], where the decoding embedding and the input embedding are shared. Since $W$ is a matrix of sense embeddings, it cannot be directly used in the decoding network for next step as in Eq. (1). Instead, we obtain embedding $e_t$ by calculating the weighted sum of sense embeddings according to their conditional probabilities. Assume that $\hat{y}_t = i$ is the input word at step $t$,

$$e_t = \sum_{j \in [1,2,...,M_i]} P(s_{t-1} = \langle i,j \rangle | \hat{y}_{[0:t-1]}) \, w_i^j, \tag{6}$$

$$P(s_{t-1} = \langle i,j \rangle | \hat{y}_{[0:t-1]}) = \frac{P(s_{t-1} = \langle i,j \rangle | \hat{y}_{[0:t-2]})}{\sum_{k \in [1,2,...,M_i]} P(s_{t-1} = \langle i,k \rangle | \hat{y}_{[0:t-2]})}. \tag{7}$$

### 4.1.2 Embedding Matching with Kernels

To calculate the probability of each sense, it is very straightforward to introduce Gaussian distributions in the embedding space. However, it is difficult to learn a Gaussian distribution for embeddings in high dimensional space for the following reasons. Context vectors are usually distributed in low dimensional manifolds embedded in a high dimensional space. Using an iostropic Gaussian distribution to model embedding vectors in low dimensional manifolds may lead to serious instability. Assume in a $d$-dimensional space, the distribution of $H_i$ follows $N(0, \sigma_1)$ in a $d_1$-dimensional subspace. We build a model $N(0, \sigma)$ to fit the embedding points. But there are often some noisy outliers, which are assumed to distribute uniformly in a cube with edge length 1 and centered at the origin. Then the average square distance between an outlier and the origin is $\frac{d}{12}$, which increases linearly with $d$. The log-likelihood to maximize can be written as:

$$\mathcal{L} = \sum_{x \in X} \log((\sqrt{2\pi}\sigma)^{-d} \exp(-\frac{\sum_{i \in 1,2,...,d} x_i^2}{2\sigma^2})) = \sum_{x \in X} (-d \log(\sqrt{2\pi}\sigma) - \frac{\sum_{i \in 1,2,...,d} x_i^2}{2\sigma^2}), \tag{8}$$

where $X$ is the set of data points including outliers. Denote the proportion of outliers in $X$ as $\alpha$. Since $\mathbb{E}(\sum_{i \in 1,2,...,d} x_i^2)$ equals $d_1$ for points generated by the oracle and $\frac{d}{12}$ for outliers, $\mathcal{L}$ is dominated by outliers when $d$ is large. The optimal $\sigma$ approximately equals to $\sqrt{\frac{\frac{\alpha d}{12} + (1-\alpha)d\sigma_1}{d}}$. With large $d$, optimal $\sigma \approx \sqrt{\frac{\alpha}{12}}$, which is independent of real variance $\sigma_1$. As expected, we find that directly modeling the Gaussian distributions does not work well in our preliminary experiments.

Therefore we design a kernel function to model embedding variances, which can be more easily learned compared with Gaussian mixture model. Specifically, we replace the inner product $\mathcal{I}(h,e) = cos(h,e)|h||e|$, which can be regarded as a fixed kernel around whole space, with a kernel function

$$\mathcal{K}_\theta(h,e) = |h| \, |e| \, (a \exp(-\theta \, cos(h,e)) - a). \tag{9}$$

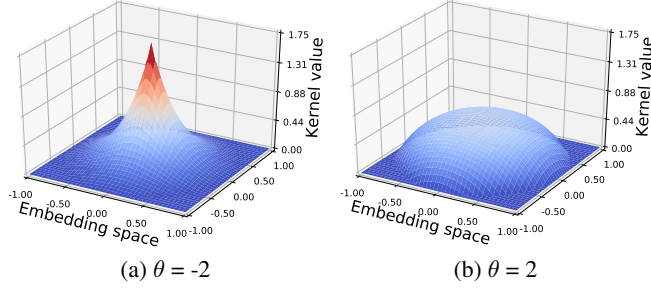

(a) $\theta = -2$        (b) $\theta = 2$

Figure 2: Kernel shapes with different $\theta$.

Here $\theta$ is a parameter controlling the embedding variances of each sense and $a = \frac{-\theta}{2(\exp(-\theta)+\theta-1))}$ is a normalization factor. When $\theta \to 0$, $\mathcal{K}_\theta(h,e) \to \mathcal{I}(h,e)$, which degenerates to common inner product. As shown in Figure 2, with a small $\theta$, embeddings are concentrated on a small region, while a large $\theta$ leads to a flat kernel. Finally, parameters for the $i$-th word could be: $\{[w_i^1, \theta_i^1], [w_i^2, \theta_i^2] \cdots [w_i^{M_i}, \theta_i^{M_i}]\}$, where $w_i^j$ and $\theta_i^j$ are the embedding and kernel parameter of sense $\langle i, j \rangle$. Intuitively, in the original space with inner product similarity, the density of probability mass is uniformly distributed. But $\mathcal{K}$ distorts the probabilistic space, making the variances of context vectors differ over different senses.

Since

$$\frac{\partial \log \mathcal{K}_\theta(h,e)}{\partial \theta} = \frac{1}{a}\frac{\partial a}{\partial \theta} - \frac{cos(h,e)exp(-\theta cos(h,e))}{\exp(-\theta cos(h,e))-1}, \tag{10}$$

the gradient of each $h$ is bounded for fixed $\theta$. It results from the continuity of $\frac{cos(h,e)exp(-\theta cos(h,e))}{\exp(\theta cos(h,e))-1}$ when $cos(h,e) \neq 0$ and the fact that $\frac{cos(h,e)exp(-\theta cos(h,e))}{\exp(\theta cos(h,e))-1} \to \frac{1}{\theta}$, when $cos(h,e) \to 0$. As a result, a small proportion of outliers or noise points will not have a major impact on training stability.

## 4.2 Training Scheme

It is difficult to empirically determine the sense numbers of each word, which is a very large set of hyper-parameters. Also, properties of the same word may vary among different corpora and tasks. So we design a training scheme for KerBS, which includes dynamic sense allocation. Instead of providing the sense number for each word, we only need to input the total sense number. The algorithm will automatically allocate senses.

Details of the training scheme are shown in Algorithm 1. Specifically, to obtain parameters for both KerBS and upstream network $f_\phi$, which outputs the context vectors, the whole process consists of allocation and adaptation phases. Before training, $W$ and $\theta$ are initialized by a random matrix and a random vector respectively. We randomly allocate $M_{sum}$ senses to words. After initialization, we first turn to the adaptation phase. Given a sequence $\hat{y}$ in training set, at step $t$, we get the context vector $h_t$ from $f_\phi$. Then sense and word probabilities are calculated by Eq. (4) and Eq. (5), respectively. Afterwards, we calculate the log-probability $\mathcal{L}$ of generating $\hat{y}_t$. And we maximize $\mathcal{L}$ by tuning $W$, $\theta$ and $\phi$:

$$\mathcal{L} = \sum_t \log(P(y_t = \hat{y}_t | \hat{y}_{[0:t-1]}; W, \theta, \phi)). \tag{11}$$

During the adaption phase, KerBS learns $w_i^j$, the sense embedding vector, and $\theta_i^j$, the indicator of distribution variance.

During the allocation phase, we remove redundant senses and reallocate them to poorly predicted words. To determine senses to remove and words which need more senses, we record the moving average of each word's log prediction accuracy $\log P$ and sense usage $U$:

$$\log P_i \leftarrow (1 - \beta) \log P_i + \log(P(y_t = i)) \mathbb{1}_{i=\hat{y}_t} \tag{12}$$

$$U_i^j \leftarrow (1 - \beta) U_i^j + \beta P(s_t = \langle i, j \rangle) \mathbb{1}_{i=\hat{y}_t} \tag{13}$$

where $\beta$ is the updating rate. For a word $i$, if after several epochs $logP$ is consistently lower than a threshold $\epsilon$, we think that the senses currently allocated to $i$ is not enough. Then we delete the least used sense and reallocate it to $i$. We alternatively perform adaption and reallocation until convergence.

---

**Algorithm 1:** Training scheme for KerBS

---

**Input** : Training corpus $\hat{Y}$, total sense num $M_{sum}$, word num $V$, embedding dimension $d$,
$\quad\quad\quad$ adaption-allocation ratio $Q$, threshold $\epsilon$;
**Output :** $W$, $\theta$, sense allocation list $L$;
Initialize $W, H, \theta, U, L, step = 0$;
**while** *not converge* **do**
$\quad$ Random select $\hat{y} \in \hat{Y}$;
$\quad$ **for** $i_t$ *in* $T$ **do**
$\quad\quad$ $h_t \leftarrow f_\phi(\hat{y}_{[0:t-1]})$ ;
$\quad\quad$ Calculate sense probability $P(y_t = \langle i, j \rangle)$ and word probability $P(y_t = i)$ by Eq. (4), (5);
$\quad\quad$ MAXIMIZE $\log(P(y_t = \hat{y}_t))$ by ADAM;
$\quad\quad$ Update $logP$ and $U$ by Eq. (12), (13);
$\quad$ **end**
$\quad$ **if** $step \bmod Q = 0$ **then**
$\quad\quad$ **for** $i$ *in* $\{1, 2, ..., V\}$ **do**
$\quad\quad\quad$ **if** $logP_i < \epsilon$ **then**
$\quad\quad\quad\quad$ $i'_0, j'_0 \leftarrow \arg\min_{i',j'}(U^{j'}_{i'})$;
$\quad\quad\quad\quad$ $\theta^{j'_0}_{i'_0} \leftarrow 1e-8$; $U^{j'_0}_{i'_0} \leftarrow$ MEAN$(U)$;
$\quad\quad\quad\quad$ $L[\langle i'_0, j'_0 \rangle] \leftarrow i$;
$\quad\quad\quad$ **end**
$\quad\quad$ **end**
$\quad$ **end**
$\quad$ $step = step + 1$;
**end**

---

## 4.3 Theoretical Analysis

In this part, we explain why KerBS has the ability to learn the complex distributions of context vectors. We only give a brief introduction to the following lemmas and leave more detailed proofs in the appendix.

**Lemma 4.1.** *KerBS has the ability to learn the multi-sense property. If the real distribution of context vectors consists of several disconnected clusters, KerBS will learn to represent as many clusters as possible.*

*Proof.* Each cluster of word $i$'s context vectors attracts $i$'s KerBS sense embeddings, in order to draw these embeddings nearer to increase $\mathcal{L}$. However, if a cluster has already been represented by a KerBS sense, its attractions to embeddings of other senses get weaker. So they will converge to other clusters. Instead of gathering together in a few clusters, senses will try to represent as many clusters of context vectors' distribution as possible. $\square$

**Lemma 4.2.** *KerBS has the ability to learn variances of embedding distribution. For distributions with larger variances, KerBS learns larger $\theta$.*

*Proof.* The optimized $\theta$ is a solution of equation $\frac{\partial \mathcal{L}}{\partial \theta} = 0$. We only need to explain that, when the variance of $h$ grows, the solution of the equation gets larger. $\square$

## 5 Experiment

In this section, we empirically validate the effectiveness of KerBS. We will first set up the experiments, and then give the experimental results in Section 5.2.

We test KerBS on several text generation tasks, including:

- Machine Translation (MT) is conducted on IWSLT'16 De→En, which contains 196k pairs of sentences for training.

- Language modeling (LM) is included to test the unconditional text generation performance. Following previous work, we use a 300k, 10k and 30k subset of One-Billion-Word Corpus for training, validating and testing, respectively.
- Dialog generation (Dialog) is also included. We employ the DailyDialog dataset from Li et al. [2017] for experiment, by deleting the overlapping of train and test sets in advance.

Note that these text generation tasks emphasize on different sides. MT is employed to test the ability of semantic transforming across bilingual corpus. LM is included to test whether KerBS can generally help generate more fluent sentences. Dialog generation even needs some prior knowledge to generate good responses, which is the most challenging task.

For LM, we use Perplexity (PPL) to test the performance. For MT and Dialog, we measure the generation quality with BLEU-4 and BLEU-1 scores [Papineni et al., 2002]. Human evaluation is also included for Dialog. During human evaluation, 3 volunteers are requested to label Dialog data containing 50 sets of sentences. Each set contains the input sentences as well as output responses generated by KerBS and baseline models. Volunteers are asked to score the responses according to their fluency and relevance to the corresponding questions. (See detailed scoring in the appendix.) After responses are labeled, we calculate the average score of each method. Then a t-test is performed to reject the hypothesis that KerBS is not better than the baseline methods.

## 5.1 Implementation Details

For LM, we use GRU language model [Chung et al., 2014] as our testbed. We try different sets of parameters, including RNN layers, hidden sizes and embedding dimensions. The model that performs best with traditional softmax is chosen as the baseline.

For MT and Dialog, we implement the attention-based sequence to sequence model (*Seq2Seq*, [Bahdanau et al., 2015]) as well as Transformer [Vaswani et al., 2017] as our baselines. For Seq2Seq, (hidden size, embedding dimension) are set to (512, 256) and (1024, 512), respectively. And For Transformer, (hidden size, embedding dim, dropout, layer num, head num) is set to (288, 507, 0.1, 5, 2) for both MT and Dialog, following Lee et al. [2018]. All models are trained on sentences with up to 80 words. We set the batch size to 128 and the beam size to 5 for decoding. For both German and English, we first tokenize sentences into tokens by Moses tokenizer [Koehn et al., 2007]. Then BPE [Sennrich et al., 2016] is applied to segment each word into subwords.

Adam [Kingma and Ba, 2014] is adopted as our optimization algorithm. We start to decay the learning rate when the loss on validation set stops to decrease. For LM, we set the initial learning rate to 1.0, and the decay rate to 0.8. For MT and Dialog, the initial learning rate is 5e-4 and the decay rate is 0.5.

## 5.2 Results of Text Generation

We list the results of using KerBS in Table 1 and 2. Then we give some analysis.

Table 1: Performance of KerBS on Seq2Seq.

| Tasks | Metrics | Seq2Seq | Seq2Seq+ MoS [Yang et al., 2018] | SeqSeq + KerBS |
|-------|---------|---------|----------------------------------|----------------|
| MT | BLEU-4 | 25.91 | 26.45 | **27.28** |
| LM | PPL | 103.12 | 102.72 | **102.17** |
| Dialog | BLEU-1 | 16.56 | 13.73 | **17.85** |
| | Human Eval. | 1.24 | 1.04 | **1.40** |

Table 2: Performance of KerBS on Transformer.

| Tasks | Metrics | Transformer | Transformer + MoS [Yang et al., 2018] | Transformer + KerBS |
|-------|---------|-------------|---------------------------------------|---------------------|
| MT | BLEU-4 | 29.61 | 28.54 | **30.90** |
| Dialog | BLEU-1 | 10.61 | 9.81 | **10.90** |

**Machine Translation** For machine translation, KerBS achieves higher BLEU-4 scores on Seq2Seq(+1.37) and Transformer(+1.29). However, the performance gain of MoS is not significant, and it is not even as good as vanilla Transformer model. Cases of MT on Transformer are shown in Table 3.

Table 3: Examples of MT on IWSLT'16 De→En

| Source | meine gebildete Mutter aber wurde Lehrerin. |
|---|---|
| Transformer | my foster mother was a teacher. |
| + MoS | and my educated mother was a teacher. |
| + KerBS | but my educated mother became a teacher. |
| Source | man erreicht niemals eine Gemeinde mit Ideen, man setzt sich mit den Einheimischen zusammen. |
| Transformer | you never achieve a community with ideas; you put together with local people. |
| + MoS | you never get a community with ideas, you're putting together with indigenous people. |
| + KerBS | you never get to a community with ideas, and you sit with the local people. |

Table 4: Examples of dialog generation on DailyDialog

| Source | what do you mean ? |
|---|---|
| Seq2Seq | i mean, what s up with the and iron bars on your windows. |
| + MoS | well, how can i put this? france is a terrible team. |
| + KerBS | well, i mean. we always do the same thing. there s no variety in our lives. |
| source | now , what seems to be the trouble ? |
| Seq2Seq | trouble is trouble. |
| + MoS | yeah. and he was. |
| + KerBS | not bad. but i have a bad cold today. |
| source | it can t be more than fourteen days late for us to accept it . |
| Seq2Seq | it will just fine. |
| + MoS | well, i see. have you been back to work then? |
| + KerBS | maybe you re right. i think we should take it. |

**Language Model** As expected, KerBS achieves lower PPL (102.17) on LM compared with both MoS (102.72) and traditional softmax (103.12). Although it introduces more parameters, KerBS does not lead to overfitting. On the contrary, the increased complexity in KerBS helps the model to better capture the information of the embedding space.

**Dialogue Generation** We also include results of dialog generation. Unlike tasks where source and target sentences are highly aligned, dialog generation may need some prior knowledge for obtaining good responses. Moreover, the multi-modality of the generated sentences is a serious problem in Dialog. We expect that much expressive structure of KerBS could help. Since the performance of Transformer is not comparable to Seq2Seq on Dialog generation, we will focus on Seq2Seq in this part. KerBS achieves a BLEU-1 score of 17.85 on test set, which is remarkable compared with the baselines. Human evaluations also confirm the effectiveness of using KerBS in dialog generation. After performing a one-tailed hypothesis test, we find that the p-value is lower than 0.05, which means that the obtained improvements on Dialog systems are nontrivial. We list some of the generated responses of different models in Table 4.

## 5.3 Ablation Study

We perform ablation study of three variants of KerBS on the MT task. KerBS w/o kernel removes the kernel function from KerBS, so that distribution variances are no longer explicitly controlled. We find that it loses 0.49 BLEU scores compared with original KerBS, which indicates that to explicitly express distribution variances of hidden states is important and KerBS works well in doing so (Table 5). KerBS with single sense replaces the multi-sense model with single-sense one, which also leads to performance decline. This further confirms our assumption that the distribution of

Table 5: Results of ablation study on MT (Seq2Seq).

| Models | BLEU-4 |
|---|---|
| Seq2Seq + KerBS | 27.28 |
| w/o kernel | 26.79 |
| w/ only single sense | 26.80 |
| w/o dynamic allocation | 27.00 |

context vectors is multi-modal. In such cases, the output layer should also be multi-modal. In KerBS w/o dynamic allocation, each word is allocated with a fixed number of senses. Though it still performs better than single sense models, it is slightly worse than full KerBS model, which shows the necessity of dynamic allocation.

### 5.4 Detailed Analysis

In this part, we verify that KerBS learns reasonable sense number $M$ and variance parameter $\theta$ by examples. And we have the following conclusions.

Table 6: Randomly selected words with different numbers of senses $M$ after training.

| Sense | 1 | 2 | 3 | 4 |
|---|---|---|---|---|
| word | Redwood heal structural theoretical rotate | particular figure during known size | open order amazing sound base | they work body power change |

Firstly, KerBS can learn the multisense property. From Table 6, we find that words with a single meaning, including some proper nouns, are allocated with only one sense. But for words with more complex meanings, such as pronouns, more senses are necessary to represent them. (In our experiment, we restrict each word's sense number between 1 and 4, in order to keep the training stable.) In addition, we find that words with 4 senses have several distinct meanings. For instance, "change" means transformation as well as small currency.

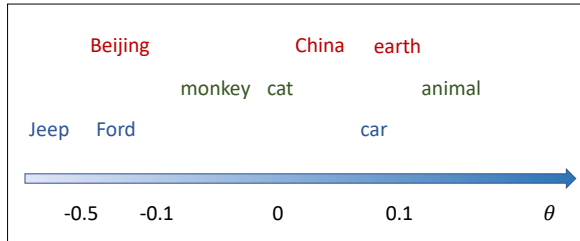

Figure 3: Words with different $\theta$.

Secondly, $\theta$ in KerBS is an indicator for words' semantic scopes. In figure 3 we compare the $\theta$ of 3 sets of nouns. For each set of them, we find words denoting bigger concepts (such as car, animal and earth) have larger $\theta$.

### 5.5 Time Complexity

Compared with baselines, the computation cost of incorporating KerBS into text generation mainly lies with the larger vocabulary for embedding matching, which is only a portion of the whole computation of text generation. Empirically, when we set the total sense number to about 3 times the vocabulary size, KerBS takes twice as long as vanilla softmax for one epoch.

## 6   Conclusion

Text generation requires a proper embedding space for words. In this paper, we proposed KerBS to learn better embeddings for text generation. Unlike traditional Softmax, KerBS includes a Bayesian composition of multi-sense embedding for words and a learnable kernel to capture the similarities between words. Incorporating KerBS into text generation could boost the performance of several text generation tasks, especially the dialog generation task. Future work includes proposing better kernels for generation and designing a meta learner to dynamically reallocate senses.

**Acknowledgments**

We would like to thank Xunpeng Huang and Yitan Li for helpful discussion and review of the first version. We also wish to thank the anonymous reviewers for their insightful comments.

## Footnotes

[1]Since there is no direct supervision, an embedding vector does not necessarily correspond to a semantic sense.

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
