[Supplementary Material · NIPS_19_Bas_Appendix.pdf]

# Appendix for Kernelized Bayesian Softmax for Text Generation

**Ning Miao   Hao Zhou   Chengqi Zhao   Wenxian Shi   Lei Li**
ByteDance AI lab
{miaoning,zhouhao.nlp,zhaochengqi.d,shiwenxian,lileilab}@bytedance.com

## A   Proofs

**Lemma A.1.** *KerBS has the ability to learn the multi-sense property. If the real distribution of context vectors is composed of several disconnected parts, KerBS components will learn to represent as many as these parts.*

*Proof.* We only prove the simplest situation under traditional inner product kernel. We assume that the real context vectors of the $i$-th word are composed of two disconnected parts and it is also allocated with two KerBS senses. We also assume that part 1 has already been represented by sense $\langle i, 1 \rangle$, i.e., $P(s = \langle i, 1 \rangle | y = i) \to 1$ for $h_1$ in part 1. Then for the second newly allocated sense $\langle i, 2 \rangle$, we find

$$\frac{\partial \mathcal{L}}{\partial w_i^2} = \sum_{h_1} \frac{\partial log(Softmax(h_1 \cdot w_i^2))}{\partial w_i^2} + \sum_{h_2} \frac{\partial log(Softmax(h_2 \cdot w_i^2))}{\partial w_i^2} \tag{1}$$

$$= \sum_{h_1} \frac{R_1 exp(h_1 \cdot w_i^2) h_1}{(exp(h_1 \cdot w_i^1) + exp(h_1 \cdot w_i^2))(exp(h_1 \cdot w_i^1) + exp(h_1 \cdot w_i^2) + R_1)} \tag{2}$$

$$+ \sum_{h_2} \frac{R_2 exp(h_2 \cdot w_i^2) h_2}{(exp(h_2 \cdot w_i^1) + exp(h_2 \cdot w_i^2))(exp(h_2 \cdot w_i^1) + exp(h_2 \cdot w_i^2) + R_2)}, \tag{3}$$

where $h_1$ and $h_2$ are context vectors in part 1 and 2, respectively. $R_i = \sum exp(h_i \cdot w_j^k)$ for all senses except $\langle i, 1 \rangle$ and $\langle i, 2 \rangle$. As part 1 has already be well represented by sense $\langle i, 1 \rangle$, $exp(h_1 \cdot w_i^1)$ should be much larger than $exp(h_1 \cdot w_i^2)$.

Then

$$\frac{exp(h_1 \cdot w_i^2)}{exp(h_1 \cdot w_i^1) + exp(h_1 \cdot w_i^2)} < \epsilon. \tag{4}$$

As a result part 1's attraction (line 2) to $w_i^2$ is much smaller than part 2 (line 3), and $w_i^2$ will move towards part 2.

$\square$

**Lemma A.2.** *KerBS has the ability to learn model variances. For distributions with larger variances, KerBS learns larger $\theta$.*

*Proof.* We will only give a heuristic proof for the situation where $\theta$ is a small positive number. The proof is also done under single-sense condition. If $\theta$ is in other intervals, the proof will be more complex, but the ideas are the same.

From the definition of $\mathcal{L}$,

$$\mathcal{L} = \sum_t \log(P(y_t = \hat{y}_t; \theta)), \tag{5}$$

where $\hat{y}_t$ is the the expected output for $y_t$, and we temporarily hide other parameters.

We can derive the partial derivative of $\mathcal{L}$ with respect to $\theta_i$:

$$\frac{\partial \mathcal{L}}{\partial \theta_i} = \sum_{t,\hat{y}_t=i} (1 - P(y_t = i)) \frac{\partial \mathcal{K}_\theta(h_t, w_i)}{\partial \theta_i} - \sum_{t,\hat{y}_t \neq i} P(y_t = i) \frac{\partial \mathcal{K}_\theta(h_t, w_i)}{\partial \theta_i}. \tag{6}$$

When $\theta$ is small, we can approximate $a$ by the following equation:

$$a = \frac{-\theta}{2(\exp(-\theta) + \theta - 1))} \approx \frac{-\theta}{2(1 - \theta + \frac{\theta^2}{2} + \theta - 1)))} = -\frac{1}{\theta}. \tag{7}$$

Approximately,

$$\mathcal{K}_\theta(h_t, w_i) \propto -\frac{1}{\theta_i}(\exp(-\theta_i cos_t) - 1), \tag{8}$$

$$\frac{\partial \mathcal{K}_\theta(h_t, w_i)}{\partial \theta_i} \propto \frac{1}{\theta_i^2}(\exp(-\theta_i cos_t) - 1) - \frac{1}{\theta_i}(-cos_t \exp(-\theta_i cos_t)), \tag{9}$$

where $cos(h_t, w_i)$ is abbreviated as $cos_t$.

Because $cos(h_t, w_i)$ is usually small for $\hat{y}_t \neq i$ we can ignore the second part of Eq. (6). So the optimal value for $\theta$ is approximately a solution to Eq. (10).

$$\sum_{t,\hat{y}_t=i} (1 - P(y_t = i)) \frac{\partial \mathcal{K}_\theta(h_t, w_i)}{\partial \theta_i} = 0. \tag{10}$$

Then,

$$F = \sum_{t,\hat{y}_t=i} (1 - P(y_t = i)) (\underbrace{exp(-\theta_i cos_t) - 1 + \theta_i cos_t exp(-\theta_i cos_t)}_{F_1}) = 0, \tag{11}$$

Hence, when $cos_t$ gets smaller, $\theta_i$ tends to increase, since $\frac{\partial F_1}{\partial cos_t} \frac{\partial F_1}{\partial \theta_i} > 0$ when $cos_t > 0$ and $cos_t$ is usually positive when $\hat{y}_t = i$. So when distribution variance increases, $cos_t$ tends to decrease, because context vectors are farther from the mean vector. As a result, $\theta_i$ will increase. $\qquad\square$

## B   Experiment Details

**Scoring Standard for Human Evaluation**   The volunteers are asked to score responses generated by all models according to the following standard:

- Score 0 : response which is neither fluent nor relative to the input question.
- Score 1 : response which is either fluent or relative to the input question, but not both.
- Score 2 : response which is both fluent and relative to the input question.