[Reviews · NeurIPS 2019]

Reviewer 1



This paper builds on the motivation that context vectors from a language model, such as BERT, often cluster into separate groups for the same next word. These clusters may correspond to different senses of the word, and often have varying variances. The authors argue that a traditional softmax is not expressive enough to capture these clusters. A similar argument was made by Yang et al in their Mixture of Softmax (MoS) paper. The solution presented here is quite different though -- to allocate multiple senses to each word in the output embedding table, and to use a parameterized kernel to model the variance. The ideas are pretty neat, and as far as i know, original. The paper is rather heavy on prose and light on results, analysis or insights. There are several intuitions presented as to why different components presented make sense, but there are no results showing that these intuitions actually make sense. Specifically, I would have liked to see at least a discussion of which words end up with higher number of senses, or higher values of \theta after training. Results are presented on 3 tasks, which is good. But all of these are on a simple baseline; it would also be interesting to see if the improvements persist with some state of the art models (e.g. using transformer architectures). There is a brief discussion about runtime, but I would like to see a more concrete comparison of the speed compared to MoS. The writing could be tightened up considerably. There is a lot of repetitive text about the intuitions behind word embeddings (e.g. lines 20-32, 77-83, 89-90, 113-118). It is not clear if how Pr(y_{t-1} = s_i^j | x_{[0:t-1}) is computed in Eq. 7 (is this the same as Eq. 5?). The x and y axes in Figure 2 are not clear, neither why the kernel presented in Eq. 9 maximize at 0? The algorithm manages to confuse the reader rather than help -- why is there a maximization step within the training loop? Why is there only a single instance of S and T being trained on? An informed reader could guess the actual intended workings of the algorithm, but scientific writing should not rely on guesses. Overall, the paper presents some compelling ideas but is lacking in results to back them up.

Reviewer 2



Originality: the approach is original and interesting. The related work is cited adequately. Quality: theoretical part seems sound but the experimental part has some flaws and can and should be improved. Clarity: paper is well written and easy to understand Significance: there is no doubt of the importance of work, e.g if authors actually demonstrated that they indeed lean multi-sense embeddings, that would be very useful. But given current state of experimentation it is unclear if the technique actually works the way authors claim.

Reviewer 3



Tackling multiple senses of words is important, and this paper makes the first attempt to resolve the problem that each word corresponds to a single (sense) vector at the projection layer in text generation. I appreciate the motivation of this work, and the proposed model improves performance on various text generation tasks. In terms of method, there's nothing particularly surprising. It adopts a sense embedding matrix instead of word embedding matrix, and uses heuristics to dynamically allocate senses to words. In addition, it employs kernels with a learnable variance in place of inner product. While these techniques are not novel, they may be practical and provide a baseline for future work.

[Author Response · NeurIPS 2019]

**Compare with the SOTA.** We carry out further experiments on Transformer and list the results in Table 1. We find that KerBS can also bring performance gains to Transformer models.

Table 1: Experimental results on Transformer.

| Method | Transformer | Transformer+MoS | Transformer+KerBS |
|--------|-------------|-----------------|-------------------|
| MT | 29.6 | 28.5 | 30.9 |
| Dialog | 10.61 | 9.81 | 10.90 |

**More analysis.** Following are some examples illustrating the effectiveness of KerBS in learning word properties.

Table 2: Randomly selected words with different numbers of senses after training.

| Sense | 1 | 2 | 3 | 4 |
|-------|---|---|---|---|
| word | Redwood heal structural theoretical rotate | particular figure during known size | open order amazing sound base | they work body power change |

Firstly, KerBS can learn the multisense property. From Table 2, we find that words with single meaning, including some proper nouns, are allocated with only one sense. But for words with more complex meanings, such as pronouns, more senses are necessary to represent them. (In our experiment, we restrict each word's sense number between 1 and 4, in order to keep the training stable.) In addition, we find that words with 4 senses have several distinct meanings. For instance, 'change' means transformation as well as small currency.

(a) Words with different $\theta$.

(b) A visualization of learned word embeddings. The radius of each sense embedding is decided by an order-preserving transformation of $\theta$.

Secondly, $\theta$ in KerBS is an indicator for words' semantic scopes. In figure (a) we compare the $\theta$ of 3 sets of nouns. For each set of them, we find words denoting bigger concepts (such as car, animal and earth) have larger $\theta$. In the pre-experiment, we build an oracle to generate word embeddings with different variances. It turns out that KerBS can accurately recover the order relation of variances between words. Please notice that in Figure (a), we use the largest $\theta$ of each sense as the $\theta$ of the word.

Figure (b) is a simple example of KerBS embedding. We find that the word 'train' has 2 senses (train_0 and train_1). While train_0 is close to the embeddings of 'practice', train_1 falls in the region of vehicles.

**Compare with MoS** Conditional text generation is more widely used than language models, but there is no experiments except LM in the paper of MoS. So we have to compare KerBS against MoS in other tasks. But the structure of MoS is kept the same, except the embedding size and the number of mixture of components. The speeds of MoS and KerBS with same number of senses are nearly equal in our experiments.

**Ablation study of dynamic allocation** We further perform a supplementary experiment on MT task. We find that with fixed sense allocation, KerBS still performs better than MoS(+0.55BLEU) and single sense KerBS(+0.20BLEU), but slightly worse than KerBS with dynamic allocation(-0.28BLEU). Moreover, the major advantage of dynamic allocation is that we can actually use a small total sense number to save computation, because only a few critical words need more than one senses. We will add a thorough analysis in the next version.

**Confusion in the Algorithm.** We are sorry for the confusion in Algorithm part and we will make it clear in the next version. (a) The computation of $P(y_{t-1} = S_i^j | x_{[0:t-1]})$ in Eq.7 is exactly the same as in Eq.5. (b) Figure 2 is a sketch map of kernel behaviours, where x-axis and y-axis is the coordinate frame of the embedding manifold, which takes a sense embedding $w_i^j$ as the origin. When the embedding of another sense is closer to the origin, their similarity gets higher. As a result, the kernel maximize at the origin. (c) The maximization in the loop is a MLE step, which maximizes the log-likelihood of generating the correct words. (d) S and T should be extracted from a corpus.

[Meta-Review · NeurIPS 2019]

This work proposes an approach to accommodate multiple senses of words as different embeddings. Reviewers' positive assessment was influenced by the additional experiments and results provided in the authors' response, so authors are definitely expected to incorporate those (or results to the same effect) in the final version.